# Optimistic Exploration in Reinforcement Learning Using Symbolic Model Estimates

**Sarath Sreedharan**
Department of Computer Science
Colorado State University
ssreedh3@colostate.edu

**Michael Katz**
IBM T.J. Watson Research Center
michael.katz1@ibm.com

## Abstract

There has been an increasing interest in using symbolic models along with reinforcement learning (RL) problems, where these coarser abstract models are used as a way to provide RL agents with higher level guidance. However, most of these works are inherently limited by their assumption of having an access to a symbolic approximation of the underlying problem. To address this issue, we introduce a new method for learning optimistic symbolic approximations of the underlying world model. We will see how these representations, coupled with fast diverse planners developed by the automated planning community, provide us with a new paradigm for optimistic exploration in sparse reward settings. We investigate the possibility of speeding up the learning process by generalizing learned model dynamics across similar actions with minimal human input. Finally, we evaluate the method, by testing it on multiple benchmark domains and compare it with other RL strategies.

Project Website: `https://optimistic-model-learn.github.io/`

## 1 Introduction

A popular trend in recent years is using symbolic planning models with reinforcement learning (RL) algorithms. Works have shown how these models could be used to provide guidance to RL agents [37, 26, 12], to provide explanations [33], and as an interface to receive guidance and advice from humans [21]. Coupled with the fact that advances in automated planning has made available a number of robust tools that RL researchers could adapt directly to their problems (cf. [11, 29, 31]), these methods have the potential to help addressing many problems faced by state-of-the-art RL methods. However, a major hurdle to using these methods is the need to access a complete and correct symbolic model of the underlying sequential-decision problems. While there have been efforts from the planning community to learn such models [19, 38], most of those methods have focused on cases where the models are synthesized from a set of plan traces, hence corresponding to the traditional offline reinforcement learning setting. Interestingly, very few works have been done in synthesizing such models in the arguably more prominent RL paradigm, namely, online RL.

To fill this gap, in this paper we propose a novel algorithm to learn relevant fragments of symbolic models in an online fashion. We show how it could be used to address one of the central problems within RL, namely effective exploration. We show how our method allows us to perform goal-directed optimistic exploration, while providing rigorous theoretical guarantees. The exploration mechanism leverages two distinct components: (a) a representation that captures the most optimistic model that is consistent with the set of observations received, and (b) the use of a fast and suboptimal diverse planner that generates multiple possible exploration paths, which are still goal-directed.

The idea of optimistic exploration is not new within the context of RL. The most prominent method being the RMax algorithm [5]. RMax modifies the reward function to develop agents that are optimistic under uncertainty. Our use of symbolic models, however, allows us to maintain an optimistic hypothesis regarding the underlying transition function. Coupled with a goal-directed planner, this lets us perform directed exploration in sparse reward settings, where we have a clear

37th Conference on Neural Information Processing Systems (NeurIPS 2023).

specification of the goal state but no intermediate rewards. As we show in this work, for a finite state deterministic MDP our method is guaranteed to generate a goal-reaching policy. Additionally, we investigate the use of a structured form of generalization rule that leverages a very simple intuition, namely the effects of an action don't depend on specific object labels but only on object types. Commonly referred to as lifted representation in planning literature, we show this rule to speed up learning with minimal human input.

The rest of the paper is structured as follows. We start with related work in Section 2. Section 3 provides a formal definition of the exact problem we are investigating and Section 5 shows the empirical evaluation of our method against a set of baselines. Finally, Section 6 concludes the paper with a discussion of the methods and possible future directions.

## 2   Related Work

As mentioned earlier, one of the foundational works in optimistic exploration in the context of reinforcement learning is R-max [5]. Even before the formulation in its current popular form, the idea of optimism under uncertainty has found several uses within the RL literature (cf. [20]). R-max can be seen as an instance of a larger class of intrinsic reward based learning [2], but one where the reward is tied to state novelty. Other forms of intrinsic rewards incentivizes the agent to learn potentially useful skills and new knowledge. A context where model simplification has been used in areas related to RL is in the context of stochasticity, where methods like certainty equivalence and hindsight optimization has been applied [4, 40]. In Section 6, we will see how we can also apply our methods directly in settings with stochastic dynamics. In regards to the user of symbolic models, the most common use is in the context of hierarchical reinforcement learning. Many works [26, 17, 37, 27], have investigated the possibility of using the symbolic model to generate potential options and then using a meta-controller to learn policies over such options. While most of these work assume that the model is in someway an approximation of the true model, all inferences performed at the symbolic level is performed over the original model provided as part of the problem. While in this work, we focused on cases where the symbolic model could in theory exactly capture the underlying model, the same techniques can also be applied to cases where the planning model may represent some abstraction of the true model. Another popular use of symbolic model is as source of reward shaping information (cf. [12]). In this context, works have also looked at symbolic models as a vehicle to precisely specify their objective [16, 13]. In terms of learning symbolic models, interestingly the work has mostly focused on learning plans or execution traces [38, 19, 6, 9].

In this context, it is worth noting and contrasting our proposed work with the ones focused on learning *safe models* [34, 19, 18, 28]. Safe models are defined to guarantee to only generate valid plans. This is an important theoretical guarantee to have if the learned model is the only source of information about the plans, or the cost of executing an invalid plan may be extremely high (e.g., because of dead-ends that might leave the AI agent stuck or if there are safety implications related to executing an incorrect action). In practice, this means learning pessimistic planning models, i.e., models that support only a subset of all possible valid plans. As we move from an offline setting to the more popular online one, such pessimism can become a burden rather than a strength. In the online setting, the agent is expected to have access to a simulator of the world, or the agent is allowed to make mistakes as part of its action execution without any irreversible damage. This means that the agent is free to try out different things until it finds a plan that works. Such methods naturally lend themselves to the use of an optimistic planning model that supports a super-set of possible plans.

In most of these works, the theoretical guarantee you are aiming for is to generate more pessimistic models that are always guaranteed to work but may overlook plausible plans. This is completely antithetical to considerations one must employ when performing explorations in common online RL settings, where the agent is either operating in a safe environment or interacting with a simulator. To the best of our knowledge, all existing online methods to acquire symbolic models [7, 24], focuses on extracting an exact representation of the true underlying model. Since our primary motivation for learning this model is to drive the exploration process, we do not have that limitation. Instead, we focus on learning a (more permissive) optimistic approximation. Also, it is worth noting that, the assumption that the system will be provided action arguments (something we will leverage to generalize learned dynamics) is one commonly made by most of these works. There are also some works that are trying to automatically acquire abstract symbolic models from an underlying MDP (including potential symbols) like that by [23]. This direction is orthogonal to our work, as symbols

produced by them may be meaningless to the human and we are explicitly trying to leverage human's intuitions about the problem.

## 3 Problem Setting

### 3.1 Deterministic MDP

The central problem we are interested in addressing is that of an RL agent trying to come up with a policy for a deterministic MDP with sparse reward. We specifically chose a setting, which forefronts the problems related to exploration, while placing less emphasis on other aspects of an RL problem (though in Section 6 how one could easily apply this approach to other settings). In particular, the underlying model (which is unknown) is assumed to be of the form $\mathcal{M} = \langle S, T, A, I, R, \gamma \rangle$. Under this definition, $S$ is a finite set that represents the set of possible states that the agent could find itself in. We will assume that this set includes a special absorbing state $\bot \in S$, allowing to capture both abnormal and normal trace termination. Additionally, there is a subset of states $S^G \subset S$, which correspond to 'goal' states, which are desirable states for the agent. $A$ is the set of possible actions and given the deterministic nature of the problem, the transition function is specified as $T : S \times A \times S \to \{1, 0\}$. We will refer to the action that transitions from states in $S^G$ to the absorbing state $\bot$ as the goal action ($a^G$). Given that we are interested in sparse reward setting, we will define the reward function as $R(s, a^G, s') = 1$ if $s \in S^G$ and $s' = \bot$ and 0 otherwise. Finally, $\gamma$ is the discount factor. Note that our work can support cases where the discount factor is equal to one.

We will refer to the transition to $\bot$ through a non goal action, as a failure transition. Now, $I \in S$ captures the initial state from which the agent starts. To enumerate the implications of the design choices we made in picking this model, consider the fact that reward is zero everywhere except for the goal. This means that any policy that can help reach the goal from the initial state, would be optimal for the agent (since the agent always starts from the initial state). Coupled with the fact that the transition function is deterministic, once the agent identifies a goal state, constructing an optimal policy is relatively easy as they can just reuse the path taken by the exploration strategy. Now, this setting also renders most existing methods that may use intermediate value bootstrapping or generalization mostly ineffective as there are no intermediate values to use. So it makes sense to focus on tabular methods as the RL baseline. In fact, possibly the only effective methods in the mainstream RL repertoire we can use are curiosity driven or intrinsic reward based methods and we will use such a method as a baseline. One of the central components we will leverage are state action traces we can sample from the underlying model. In particular, we say that a trace $\langle s_0, a_1, .., a_i, s_i, .., a_{k+1}, s_{k+1} \rangle$ is valid or equivalently goal-reaching if $s_0 = I, s_k \in S_G, s_{k+1} = \bot$, and for every $0 \leq i \leq k$ we have $T(s_i, a_{i+1}, s_{i+1}) > 0$. The action $a_{k+1}$ is the goal action.

In the course of discussion, we will use the word 'original model' to refer to this true but unknown underlying MDP. The agent itself is expected to be either interacting with a generative simulator that encodes this MDP or is acting in the true environment provided that they can reset to the initial state at the end of each episode.

### 3.2 Symbolic Planning Models

For the symbolic model, we will be using an a representation scheme called Planning Domain Definition Language or PDDL. In particular, we will consider a version that will ignore object types [14]. Here, a planning task is defined in relational terms, i.e., states are described in terms of objects and relationships between objects and each action is described in terms of the objects involved in that action and how they may affect or be affected by the relationships between these objects. Such a model is usually defined by the tuple $\mathcal{M}^S = \langle \mathcal{L}, \mathcal{O}, I, G \rangle$, where $\mathcal{L}$ is a first-order language, $\mathcal{O}$ a finite set of action schemas, $I$ and $G$ are specifications of the initial state and the goal, respectively. The first order language describes the objects and the relationships between these objects (captured as *predicates*). Additionally, first order language allows specifying predicates over variables as well as actual objects. Formally, the first order language is specified as $\mathcal{L} = \langle \mathcal{B}, \mathcal{V}, \mathcal{P} \rangle$, where $\mathcal{B}$ is the set of all objects, $\mathcal{V}$ are the variable names and $\mathcal{P}$ are the predicates. Each predicate $p \in \mathcal{P}$, will take a fixed number of arguments. For the purpose of discussion in this paper, we will either have cases where the arguments consist of only variables or only objects. We will refer to the former case as being the lifted representation of the predicate and the latter as a grounded instance of the predicate (or ground predicate). In general, however, predicates can be partially grounded, with some of the arguments being actual objects while others being variables. States of the model correspond to truth assignments to ground predicates. Each ground predicate can take either a true or a false value. Each

possible state for a given model is captured by a specific instantiation of all ground predicates. Thus, possible states can be uniquely represented by sets of ground predicates that are true (assuming the rest to be false under the closed world assumption). In this representation, $I$ denotes a set of ground predicates, capturing the unique initial state. For our purposes, $G$ will be captured by a subset of ground predicates, denoting all states where all of these ground predicates (and possibly some more) are true. All such states will be considered valid goal states.

Each action schema $o \in \mathcal{O}$ provides the basic structure shared by a set of ground actions that can be actually executed by the agent. Each action schema is defined as $o = \langle params(o), pre(o), add(o), del(o) \rangle$, where $params(o)$ indicates the parameters of the action schema (variables and objects), the preconditions $pre(o)$, the add-effects $add(o)$, and the delete-effects $del(o)$. The latter three are first-order formula over the language $\mathcal{L}$, specifying the conditions that must be satisfied for the action to be executable in a state, as well as the change in a state resulting from executing the action. Ground actions are obtained from the action schema by assigning objects to variables in the parameters. The agents are executing the ground actions and therefore it is common to describe the semantics of ground actions, henceforth simply referred to as actions. In this work we restrict ourselves to preconditions in disjunctive normal form. For ground actions these would be disjunctions of conjunctions of ground predicates. All states in which the formula holds, the action is applicable. The add/delete effects are conjunctions of ground predicates, making these predicates true respectively false in the state resulting in successful action execution.

For a given action schema $o$, we will denote the grounded instance obtained by replacing the parameter with an object list $\Theta$ using the function symbol $\Gamma^{\uparrow}(o, \Theta)$. We can also define an inverse mapping function $\lambda^{\uparrow}(o^{\downarrow}, \Theta)$ that retrieves the lifted model given a grounded instance (we can do this by replacing all instances of an object with a variable). This lifting function $\lambda^{\uparrow}$ is well defined in all cases where we don't have a repeating object in $\Theta$. In this particular work, we will only focus on applying such lifting functions in cases where they are well defined. Overloading the notations a bit, we will also apply the functions $\Gamma^{\uparrow}$ and $\lambda^{\uparrow}$, to create grounded and lifted instances of predicates as well. Each planning problem can be represented equivalently in a grounded form as $\mathcal{M}_{\downarrow} = \langle F_{\downarrow}, A_{\downarrow}, I, G \rangle$, where $F_{\downarrow}$ consists of all grounded predicates and $A_{\downarrow}$ grounded actions. At most this model may have $2^{|F_{\downarrow}|}$ states. A solution for a planning model is a plan $\pi = \langle a_1, ..., a_n \rangle$, which is a sequence of action whose execution in initial state will lead to a goal state, i.e., $\pi(I) \supseteq G$ (where $\pi(I) = a_n(....a_1(I))$).

### 3.3  Connecting the Symbolic Model to the MDP

For any given deterministic MDP $\mathcal{M}$ of the form defined in Section 3.1, there must exist a symbolic model that can exactly capture the MDP. In particular, there is a surjective function (many-to-one) mapping the (ground) actions in the symbolic model to MDP actions. Every plan under the symbolic model maps by this mapping to a valid trace of the MDP. The appendix include a proof that shows by construction how such a model will always exist. However, rather than creating an arbitrary mapping to a symbolic model, we are interested in creating one that leverages the expertise of a human domain expert to creating a potentially more effective representation of the problem. In particular, we start by taking human input to learn how to symbolically represent the states of the MDP. In particular, we expect the human to specify a set of predicates and objects that they might associate with the given problem. We use the symbol $F_{\downarrow}^{\mathcal{C}}$ to capture the set of all ground predicate possible under this specification. Similar to previous explanatory works [33, 32, 21] that have tried to learn symbolic representations of RL problems, we use this to learn binary classifier that test whether a ground predicate may be true in a given MDP state. We can learn such classifiers by collecting positive and negative examples for each ground concept. Once the classifiers are available, we can construct the symbolic state corresponding to each MDP state, by testing each classifier on any given MDP state. We use the function $\mathcal{C} : S \rightarrow 2^{F_{\downarrow}^{\mathcal{C}}}$ as a way to capture the mapping between the states. For potential actions, we assume that every symbolic ground action corresponds to exactly one action in the MDP. Overloading the notations a bit, we use $\mathcal{C}^{-1}(a)$ to represent the MDP action corresponding to the symbolic action $a$. As we will see later, the agent can also potentially leverage the human's intuitions about how they structure actions to further improve the effectiveness of our method. Finally, we expect the human to provide a specification of the goal states specified in terms of the ground predicates in $F_{\downarrow}^{\mathcal{C}}$. We denote this goal specification by $G^{\mathcal{C}}$. Additionally, we require that the initial state for the symbolic model corresponds to $\mathcal{C}(I)$ and for any goal state $s \in S_G$, $\mathcal{C}(s)$ satisfies $G^{\mathcal{C}}$ (or, equivalently, there is a symbolic goal action whose precondition meets this requirement) .

# 4 Our Approach

**Algorithm 1** Iteratively refine the model until a goal reaching trace is found

---

Iterative-Model-Refinement
*Input*: $\mathcal{M}_0^{\mathcal{C}}, \kappa$
*Output*: An action sequence $\langle a_1, ..., a_k \rangle$ that will lead to the goal
*Procedure*:
$\mathcal{M}_{curr} \leftarrow \mathcal{M}_0^{\mathcal{C}}$
execution_statistics $\leftarrow \{\}$, solvability_flag $\leftarrow$ True
**while** solvability_flag is True **do**
  $\widehat{\mathcal{M}}_{curr} \leftarrow$ PruneModel($\mathcal{M}_{curr}$, execution_statistics)
  $\widehat{\Pi} \leftarrow$ DiversePlanner($\widehat{\mathcal{M}}_{curr}, \kappa$)
  **if** $|\widehat{\Pi}| > 0$ **then**
    **for** $\widehat{\pi} \in \widehat{\Pi}$ **do**
      **if** $\widehat{\pi}$ leads to goal in the environment **then**
        **return** $\widehat{\pi}$
      **else**
        $\mathcal{M}_{curr}$, execution_statistics $\leftarrow$ UpdateModel($\mathcal{M}_{curr}, \widehat{\pi}$, execution_statistics)
      **end if**
    **end for**
  **else**
    solvability_flag $\leftarrow$ False
  **end if**
**end while**
**return** No policy with non-zero Value

---

The basis of our approach is an observation that every deterministic MDP has a *precise* symbolic representation. By precise representation, we mean that for the specific setting we consider here, there exists a symbolic model that can exactly simulate the MDP: any transition possible under the symbolic model must correspond to non-zero probability transition possible under the MDP and vice versa. However, as discussed, our objective is not to learn such a precise representation but rather only to learn an optimistic approximation. We start from a trivially optimistic representation of the underlying model, which we iteratively refine towards the true representation. At each iteration, the current symbolic representation is used to generate potential plans to the goal. These plans are then tested out in the environment and the observed outcomes of the execution of such action sequences are then used to refine our estimate towards the true representation. At every point of our model refinement process, we ensure that every subsequent model estimate generated is an optimistic one. By maintaining the optimistic nature of the representation, we ensure that no potential valid solution is overlooked at any point in the learning process. So we will start the discussion of our approach by providing a rigorous definition of what we mean by an optimistic representation. In particular, we are interested in creating symbolic representations that allow all valid traces that are possible under the original MDP to be possible under the new representations. Formally, we define this requirement as

**Definition 1** *For an MDP model $\mathcal{M}$, a symbolic model $\mathcal{M}^{\mathcal{C}}$ defined over a symbol mapping $\mathcal{C}(\cdot)$ is said to be an **optimistic representation**, if for every action sequence $\langle a_1, ..., a_k \rangle$ such that there exists a valid trace (i.e. it reaches goal), there exists a valid plan in $\mathcal{M}^{\mathcal{C}}$ of the form $\pi = \langle a_1^{'}, ..., a_k^{'} \rangle$, such that $\mathcal{C}^{-1}(a_i^{'}) = a_i$.*

For the given set of grounded actions $A_{\downarrow}^{\mathcal{C}}$ and a grounded set of predicates $F_{\downarrow}^{\mathcal{C}}$, we can create a symbolic model that is guaranteed to be optimistic for any MDP whose action set is isomorphic to $A_{\downarrow}^{\mathcal{C}}$ and the state space can be represented using $F_{\downarrow}^{\mathcal{C}}$. In particular, the model will have empty preconditions and delete effect and the add effects would correspond to the set of all ground predicates. This means that every action is executable in every state and an execution of any action will satisfy the goal. We will denote this model as $\mathcal{M}_0^{\mathcal{C}} = \langle F_{\downarrow}^{\mathcal{C}}, A_{\downarrow}^{\mathcal{C}}, I^{\mathcal{C}}, G^{\mathcal{C}} \rangle$. More formally, every action $a \in A_{\downarrow}^{\mathcal{C}}$ will be defined as follows: $a = \langle pre_0^a, add_0^a, del_0^a \rangle$, where $pre_0^a = del_0^a = \emptyset$ and $add_0^a = F_{\downarrow}^{\mathcal{C}}$. The fact that its an optimistic representation for any MDP possible in this context can be trivially proved (Proof is provided in the appendix).

## 4.1 Refining the Model

Now, of course, while all valid traces for the original model correspond to a plan in $\mathcal{M}_0^{\mathcal{C}}$, the symbolic model may also support plans that may not correspond to any valid trace in the original model. Our basic strategy would be to use this model as a starting point to sample potential plans, simulate/execute them in the environment or simulator and use the outcomes (both successful and failed executions) to refine the current the current estimate. We will continue this process until we find a plan that leads us to the goal. Keeping this general approach in mind, the next step would be to define our model update rule. In particular, let us assume that we receive the following observation from the environment $\langle s, a, s' \rangle$, such that $s' \neq \perp$. Now we know this corresponds to the symbolic observation

$\langle \mathcal{C}(s), \mathcal{C}(a), \mathcal{C}(s^{'}) \rangle$. Given this observation, we know that any changes made in the state must be the result of the action. We will use this information to update action's effects. For add effects, if the estimate previously had hypothesized the action making a predicate true, which doesn't hold in $\mathcal{C}(s^{'})$ then it can be removed from the add effects. Similarly, if there was a predicate that is made false in $\mathcal{C}(s^{'})$ but was not part of the delete effects, it can be added to the set of delete effects. Formally, we can set the new estimate of the action as follows $a = \langle pre_{i+1}^a, add_{i+1}^a, del_{i+1}^a \rangle$, where $pre_{i+1}^a = \{\phi | \phi \in pre_i^a \text{ and } \phi \subseteq \mathcal{C}(s)\}$ and for effects we have $add_{i+1}^a = add_i^a \setminus (F_{\downarrow}^{\mathcal{C}} \setminus \mathcal{C}(s^{'}))$ and $del_{i+1}^a = del_i^a \cup (\mathcal{C}(s^{'}) \setminus \mathcal{C}(s))$. If the sampled transition corresponds to a failure ($\langle s, a, \perp \rangle$), we will only update the precondition. Specifically, we will remove any precondition clause that satisfies the state and replace it with a set of preconditions that includes one of the predicates that was false in the model (this follows from the fact that the action failed because some predicate part of the true underlying precondition wasn't true in the given state). More formally, for any $\phi \in pre_i^a$, such that $\phi \subseteq \mathcal{C}(s)$, we remove $\phi$ and add $\Phi = \{\phi \cup f | f \in (F_{\downarrow}^{\mathcal{C}} \setminus \mathcal{C}(s))\}$. The proof for why this update rules result in optimistic representations are provided in the appendix.

## 4.2 Overall Algorithm

Algorithm 1 presents the overall iterative algorithm we will be using to identify the action sequence that can lead to a goal state. The algorithm starts with the initial estimate of the model. It iteratively generates plans for the model estimate, which will then be used to progressively refining the model until we get a plan that corresponds to a path to a goal state. These plans are derived using a diverse planner that identifies a set of plans that are diverse in terms of the actions used. This is represented by the procedure *DiversePlanner* that takes the number of diverse plans to be generated as an argument ($\kappa$). Readers can check [22] for a more detailed discussion of diverse planners. These plans are first tested on the underlying environment/simulator to check whether they lead to the goal from the initial state and if not the experiences sampled from their execution are used to refine the current model. Note that, given the optimistic nature of the model estimate, the planner would generally try to use actions that haven't been previously executed successfully. However, each future use of the action would become progressively harder due to the growing precondition set. With that said, one could further improve the planner behavior by being more careful about the actions being used as part of plans. If an action has been tested quite frequently, it would be better to de-prioritize its usage until no better alternative has been found. Note that this is quite similar to the kind of exploration performed in the context of multi-armed bandits [25]. In fact, one could directly apply methods like UCB [3] to select the action sets to be considered by the planner. This part of the algorithm is captured by the procedure *PruneModel*. To keep our implementation of the approach simple, we will use a simple queue based system to identify the actions to be included. The exact procedure we use to control the selection of actions is described in the appendix. The variable *execution_statistics* keeps track of previous action trials and the frequency of success per action. The procedure *UpdateModel* uses the rules described in Section 4.1 to use the sampled traces to update the given model estimate. One could also further improve the efficiency of the search by always testing all possible actions in every new state that is identified as part of the procedure.

**Theorem 1** *Algorithm 1 will (a) terminate in a finite number of steps and (b) identify a path to a goal (provided one exists); as long as the diverse planner used is complete (i.e., it will return a non-empty plan set as long as there exists a valid plan).*

The proof for the theorem is provided in the appendix.

**Leveraging the Exploration Strategy in the Context of RL algorithms:** In the context of the overall RL learning process, this exploration method will be used as a way to update the Q values (and depending on the algorithm, structures like replay buffers). Specifically, we will first run this exploration procedure to find a valid trace to the goal. Once such a trace is found, we can update the Q values of all the states that are part of that trace to a more informed value. Once updated, we can employ traditional RL algorithms to identify optimal policies. One could also leverage the proposed method in conjunction with other exploration strategies, during the learning process. It is important to note that any consecutive use of our approach for generating goal directed paths would be much more efficient, as the method will start from a more refined estimate of the model.

**Leveraging Lifted Representation** The algorithm described above tests each of the available actions to learn a symbolic model corresponding to the observed behavior. However, one of the important points to note here is the fact that this means that the testing and by extension learning of the

model occurs at the level of ground actions. As we had discussed earlier, a very common assumption made throughout symbolic models is that of the existence of a lifted representation of actions. Namely, the fact that the nature of actions could be described independently of the exact objects it may be interacting with. This is a very natural outcome that comes out of relational representations of tasks, where the state is represented in terms of objects and relationship between objects. Consider a simple domain where a robot is tasked with stacking blocks on the table (popularly called blocksworld [35]). It is very easy to see that the outcome of picking up the red block should be quite similar to the case of picking up the green block of the same size. For example, if we observe that the execution of the action 'pick-up red block' results in the agent holding the red block in it's gripper; then it would be quite natural to assume that the execution of 'pick-up green block' should result in the agent holding the green block. We will leverage such symmetry by asking the human to provide some additional information about each action. Specifically, the human can provide us a basic annotation over what actions could share a lifted structure and what objects each actions might interact with. Note that we are not asking the users to specify what the lifted structure may be, but just a grouping of actions and an ordered list of relevant objects. The order may reflect the different roles played by the objects participating in the action. For example, when an object is being placed on top of another, the annotation may list the destination object first and then the object being placed on top of it. The exact ordering wouldn't matter provided they remain consistent through the annotations. Additionally, even if the grouping provided by the human may be a subset of the true possible grouping and the human provides a superset of the objects relevant to any given action, our generalization approach remains valid. The set of objects associated with each action could also be automatically extracted from natural language descriptions of actions, as performed by works like that by [10].

For a given set of actions that are marked as being grounded instances of the same lifted action, we will ensure that learned effects of all actions comply with the most refined action in the set. As discussed earlier, the effects of an action comprises of add and delete effects and for each component we can select the most refined set independent of each other. From the set of effect descriptions, we select the add effect set containing the minimum number of elements and the delete effect set containing the maximum number of elements. For each such set, we can create the lifted description using the $\lambda^\uparrow$ function described earlier. Let min_add be the lifted description corresponding to the smallest set of adds and max_del be the largest set of deletes for a given set of actions corresponding to the same lifted action. Then we can simply replace the effect of every action with a grounding of these lifted actions. This will still result in an optimistic model description, as we can show that the min_add and max_del are still optimistic estimates

**Proposition 1** *Let $\bar{A} = \langle a_1, ..., a_m \rangle$ be a set of actions marked as being instances of a single lifted action $a^\uparrow$. Then* min_add *must be a superset of add effects of $a^\uparrow$ and* max_del *a superset of deletes of $a^\uparrow$, where* min_add *and* max_del *are calculated for $\bar{A}$*

The proof for validity of this proposition is discussed in the appendix. This proposition now means that, once max_del and min_add are identified, then for every action $a$ in the set of possible groundings we can replace add effects and delete effects with the corresponding grounding of the lifted effects, i.e, $add^a = \Gamma^\uparrow(\text{min\_add}, \Theta^a)$ and $del^a = \Gamma^\uparrow(\text{max\_del}, \Theta^a)$, where $\Theta^a$ is the object list corresponding to the action $a$. One can follow similar lines of reasoning to show that the lifted description of the maximal precondition description is guaranteed to entail the true preconditions.

**Lifted Representation as a Basis for Curriculum Learning.** While the above discussion focused on leveraging lifted representation to speed up learning within a given planning problem, one could also use lifted representations as basis of transferring model information from one problem instance to another. Within classical planning problems, it is very common to separate the domain information represented in lifted terms from specific problem instances. Each problem instance could differ in terms of the number and identities of objects involved, the initial and goal state. However, actions applicable in every instance share the same lifted definition. Even within RL, benchmarks consisting of various instances of the same problem domain are becoming more popular (Minigrid [8] being a popular example). When such a suite of problem instances are available, one could further speed up learning by using the smaller instance to learn as much of the lifted model as possible. Once such a partial lifted model is learned, it can then be used to refine the optimistic model in the target problem, where the normal learning process then takes over. Note that the access to a smaller problem instance doesn't obviate the need to perform learning in the true underlying model. After all, in the smaller problem there may be lifted actions that may not be executable in any of the reachable states, but needs to be used in the target problem to reach the goal.

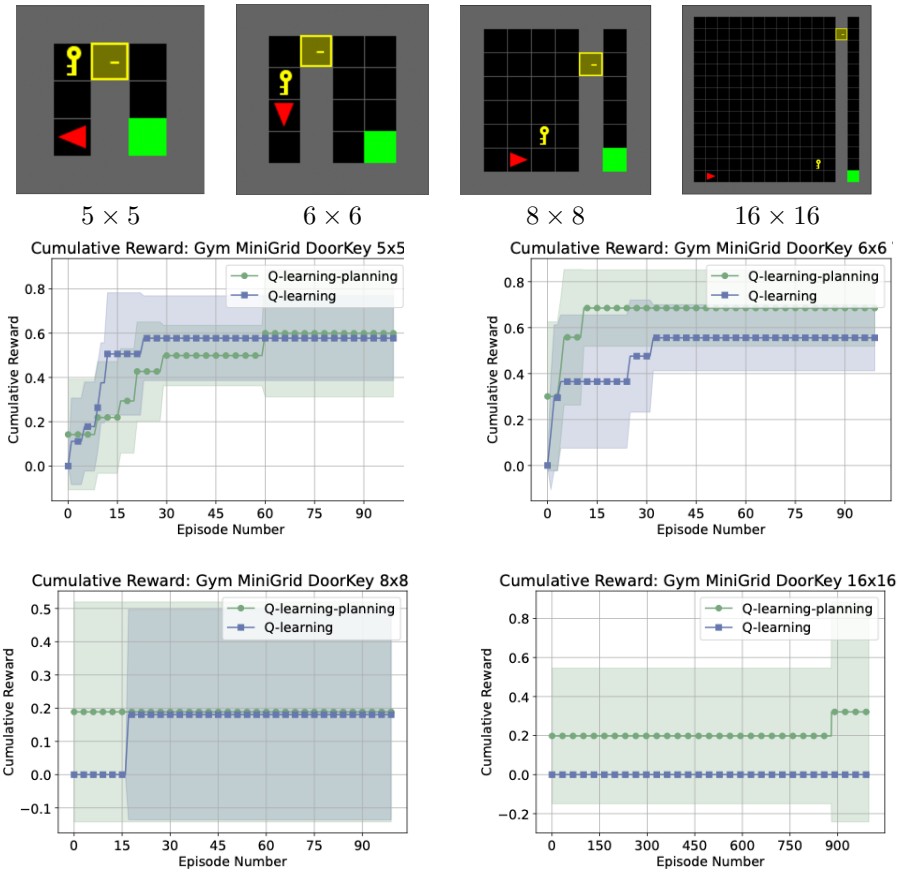

Figure 1: Four minigrid maps considered and the cumulative rewards per episode comparing our method to a vanilla RL. The values are plotted along with their 95% confidence interval.

## 5  Evaluation

We perform our evaluation in four different domains. Three of these correspond to traditional planning domains and one a more traditional reinforcement learning benchmark. The planning benchmarks include blocksworld, a simple gridworld type domain involving robot picking up objects and a domain where the agent has to control elevator schedules. For the RL domain, we looked at two variants of minigrid problem. One was the version introduced by [26] (henceforth referred to as Minigrid-Parl) and the other being a simplified version of the original minigrid testbed [8]. We chose to use the Minigrid-Parl, since it provided us with annotations that allowed us to use hierarchical RL methods. For the latter variant, we dropped the turn action and introduced directional movement, pick up, drop and toggle actions. This allowed us to use simpler PDDL formalisms to capture the domain. For each planning domain, we selected five different problems (the sizes are approximately listed in the tables in terms of the number of grounded predicates) and two problems for the minigrid domain. We created a simulator wrapper around PDDL models for each of the problems, as it allowed us easy access to the annotation information for lifting. For the minigrid problems, we auto generated PDDL problem files from the simulator code for each specific environment. he code can be found at `https://github.com/sarathsreedharan/ModelLearner`.

### 5.1  Reaching Goal States

As a first step, we are interested in testing how our proposed method compares against standard exploration strategies used by RL algorithms. In particular, we compare our method against three baselines: vanilla $\epsilon$-greedy exploration (as implemented by the SimpleRL framework [1], as part of the Q learning agent), R-max based exploration strategy (again taken from the SimpleRL framework), which as we discussed is a form of intrinsic reward, and a hierarchical RL method that learns a policy over SMDP using PPO (as implemented by [26]) on instances from Minigrid-PARL.

| Problem Instance | | ε−greedy | | | Our method (w/o lifting) | | | Our method (with lifting) | | |
|---|---|---|---|---|---|---|---|---|---|---|
| Name | Size | solved | time | # samples | solved | time | # samples | solved | time | # samples |
| Blocks | 25 | | 0.89 | 9164.2 | | 26.7 | 19262 | | 5.59 | 592 |
| | 25 | | 11.35 | 115136.8 | | 399.35 | 168859.6 | | 31.96 | 56404.8 |
| | 25 | 3/5 | 1.99 | 20702 | 3/5 | 46.86 | 18901.8 | 5/5 | 9.28 | 4432.4 |
| | 36 | | - | 2966996.4 | | - | 96152.8 | | 32.99 | 191451.2 |
| | 36 | | - | 3135384 | | - | 142605.4 | | 33.93 | 138203.4 |
| Elevator | 20 | | - | 2045316.4 | | 408.79 | 3394856.8 | | 36.94 | 88108 |
| | 20 | | - | 2087898.2 | | - | 3839855.6 | | 26.73 | 66507 |
| | 20 | 0/5 | - | 2062441.2 | 2/5 | 401.85 | 3053364.2 | 5/5 | 21.25 | 88835 |
| | 20 | | - | 2089477.6 | | - | 3099277.6 | | 36.12 | 83296 |
| | 20 | | - | 2081322 | | - | 4117586.6 | | 36.57 | 87747.2 |
| Gripper | 25 | | 5.56 | 53929.8 | | 73.52 | 77450.2 | | 15.9 | 16523 |
| | 25 | | - | 1511944.6 | | 328.74 | 252954.4 | | 23.17 | 308598.6 |
| | 25 | 1/5 | - | 2663695 | 2/5 | - | 83551.8 | 5/5 | 35.93 | 309964.8 |
| | 36 | | - | 2395190.2 | | - | 55209.8 | | 43.31 | 308598.6 |
| | 36 | | - | 2112899.6 | | - | 37136.8 | | 61.99 | 624489.6 |
| Minigrid-PARL | 94 | 0/2 | - | 2743179.4 | 0/2 | - | 310032 | 1/2 | 86.41 | 342981.8 |
| | 593 | | - | 334535.2 | | - | 2458.2 | | - | 8217319.4 |

Table 1: Our method w/wo lifting generalization vs. Q learning. Times are listed in seconds and only the average time and number of samples are reported (full data is in the Appendix).

Our interest is not only to see how well the current method performs, but also to see how much is contributed by the action-level generalization provided by lifted representations. Our primary metrics of evaluation are going to be, (a) do the methods consistently reach the goals, (b) the number of samples collected from the environment as part of reaching the goal, and (c) the time taken by the method to reach the goal. This third aspect is an important one to consider to make sure that the RL based exploration is given a fair chance when compared against planning based methods. After all, the planning methods reason over environment model, allowing them to perform less interactions with the environment. However, this adds a computational overhead, that might not be required for other method, such as vanilla RL methods. We capture that tradeoff of one computation for another by measuring the time to reaching the goal. Additionally, we set a time limit on the exploration step, as for some of these problems the exploration might not be completed in a reasonable amount of time. For all planning based instances we set the time limit to 10 minutes, while for the minigrid instances we extended the time limit to 30 minutes. Every experiment is run five times, averaging the results to account for possible randomness in the learning process. All seed values were randomly assigned and kept constant through the all five runs. As the underlying diverse planner, we used FI [15], generating ten different plans at every step. Table 1 presents the comparison of our method against Q learning for the planning benchmarks. Both R-max and SMDP time out on all tested instances, so we will skip reporting their values in the table. SMDP took 188416.8 and 106821.4 samples each for the two minigrid problems. We see that apart from Blocksworld and minigrid domain, our vanilla method is able to solve more problems and our method equipped with the application of lifting rule outperforms both by a wide margin. Neither R-Max or SMDP visited any of the goal state in the given time limit.

## 5.2 Overall Learning Performance

With the initial results collected on how well our method is able reach goal. The next question we wanted to answer was how well an RL algorithm augmented with our new exploration strategy is able to perform. For this question we focused on tabular Q learning and the minigrid environment. Specifically, we compared an instance of Q learning algorithm where Q values were initialized using the plan generated through our exploration process and a vanilla one that started with no such information ( Top row of Figure 1 provides a visualization of the maps considered for these experiments). Columns two and three of Table 2 presents the time and samples taken to get such a plan for each of the problem instance considered. The bottom two rows of Figure 1 present how the total value per episode changes over episodes. As expected, the access to a valid plan (which isn't necessarily optimal) ensures that our method starts from a higher value. For smaller instances, we see that the vanilla method eventually catches up or at least gets closer. However for the largest problem instance, even after 1000 episodes the RL agent is still unable to get a positive reward since it never reaches a goal state.

| Problem | with original *PruneModel* | | with bootstrapping | | w/o bootstrapping | |
|---|---|---|---|---|---|---|
| | time | # samples | time | # samples | time | # samples |
| DoorKey-$5 \times 5$ | 35.39 | 23977.2 | N/A | N/A | N/A | N/A |
| DoorKey-6x6 | 34.45 | 24418.4 | 42.14 | 27666.0 | 35.91 | 21531.8 |
| DoorKey-8x8 | 54.63 | 260839.8 | 52.1 | 143223.4 | 92.78 | 367749.2 |
| DoorKey-16x16 | 1488.08 | 6705823.2 | 2353.63 | 11088198 | – | – |

Table 2: The results from our method (with lifting) applied to the second Minigrid variant. Includes the performance under our original *PruneModel* strategy and the results on how bootstrapping helps learning under the new action selection strategy. The entries for DoorKey-16x16 are skipped for w/o bootstrapping, since the method timed out after 90 minutes.

## 5.3 Curriculum Learning

The final question we were interested in investigating was whether having access to a smaller problem instance had the potential of speeding up our learning process. We again went back to the minigrid problems and tested whether first learning a partial lifted model using the a problem of grid size $5 \times 5$, help speed up learning in larger problems. In particular, we stop learning in the smaller instance, as soon as we find a single solvable plan. We then use the lifted model information learned from the smaller instance to create a more informed model of the target problem. Then we follow the same procedure as before, until all the plans returned by the diverse planner is valid. One of the things we noticed was that bootstrapped models did significantly worse off under the original *PruneModel* method we used. We noticed that using a strategy based on failure count resulted in useful actions getting tested early and getting removed from consideration. So for this set of experiments we considered a different strategy that re-introduces actions more frequently. Last four columns of 2, presents the results for the minigrid problems, where the problem were bootstrapped with a lifted model learned from a $5 \times 5$ grid. The method without bootstrapping timed out for the largest problem without finding a solution after 90 mins. Except for the smallest problem we see the bootstrapping giving a definite advantage.

## 6 Conclusions and Discussion

Our paper, presents a novel exploration paradigm for reinforcement learning algorithms. Our proposed method supports the learning and refinement of optimistic symbolic estimates of the underlying model for the given task. We show how we can start with a trivially optimistic model and then use a diverse planner to drive both the task-level exploration and the refinement of the model. Experiences generated from the execution of identified plans lead to better estimates of the task, which in turn leads to more informed plans. We additionally show how we can leverage lifted representations of the given task, to generalize any learned model information across various instances of the same lifted operators. We also use this mechanism to propose a novel curriculum learning paradigm for model learning. The effectiveness of our proposed method depends on three crucial factors: (a) the possibility of performing systematic refinements of our models while ensuring desirable properties, (b) the availability of fast, diverse planners, and (c) the ability to leverage human intuition about the task. The latter is of crucial importance: even if there were other model classes and planners we could exploit, the ability to tap into human knowledge gives us a significant advantage. Importantly, the same knowledge has been used by many of the other state-of-the-art methods. Further, it only represents a small subset of the information usually provided as part of a complete symbolic planning model. One of the aspects not discussed in the paper was the fact that instead of starting with an empty model, we could have started with a partially complete model. In such cases, the human could just provide whatever they know about the task, and the RL agent can fill in the rest. We expect such settings to provide even more advantages to our method. For future work, a promising direction is to support stochastic transitions. One possible way of using such methods in a stochastic setting would be by considering a separate copy of an action for each possible transition, similar to the methods used by many probabilistic planners [39]. The central challenge here would be to recognize the different transitions associated with the same action and to ensure that the estimate at any given point is still an optimistic estimate of the true model. Last, but by no means least, is the combination with RL methods that use function approximation, especially in settings where the symbolic model might be an abstraction of the true underlying model. Such settings are among the most practical ones from the real-world applications perspective, allowing our method to gradually generalize to fine-grained abstractions and eventually to the real world.

## Acknowledgement

Sarath Sreedharan's research is supported in part by grant NSF 2303019.

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

## A    Appendix Overview

In this appendix, Section B will cover the formal statements and proof sketches for various theoretical results, Section C will cover the implementation details including hyperparameters. Section E will cover the various assumptions made in the work and finally Section F provides an overview of how our approach is used in the context of Q learning.

## B    Theoretical Results

First result we are interested in establishing is the fact that for any given MDP of the form described in Section 3.1 in the main paper, there exists a corresponding symbolic model that meets the criteria discussed in Section 3.2 in the paper.

**Proposition 2** *For an MDP of the form* $\mathcal{M} = \langle S, T, A, I, R \rangle$, *there exists a grounded symbolic model* $\mathcal{M}_\downarrow = \langle F_\downarrow, A_\downarrow, I, G \rangle$, *such that there exists*

   1. *a mapping* $\mathcal{C}$ *from the state* $S$ *of* $\mathcal{M}$ *to the states of* $\mathcal{M}_\downarrow$,

   2. *a mapping* $\mathcal{C}^{-1}$ *from the actions* $A_\downarrow$ *of* $\mathcal{M}_\downarrow$ *to the actions* $A$ *of* $\mathcal{M}$, *and*

   3. *a mapping from valid traces in* $\mathcal{M}$ *to valid plans in* $\mathcal{M}_\downarrow$ *and vice-versa.*

**Proof Sketch:**  We can build such a model by adding one grounded predicate (each corresponding to a unique lifted predicate of arity 0) for each state other than $\perp$ into $F_\downarrow$. Now a state in $S$ maps (defined by $\mathcal{C}$) exactly to the symbolic state where the corresponding predicate is true and none of the other fluents are true. Now for the action, we will start with an action definition that includes conditional effect, and then convert it to a form assumed by the work. A conditional effect captures cases where an effect of an action only fires if the state meets certain criteria. Now we create one symbolic action for each MDP action. For each possible transition between states (other than $\perp$), we will add a corresponding conditional effect that takes the predicate corresponding to source state as condition and as effect the predicate corresponding to the target state. We will keep as precondition of the actions a disjunctive list of all possible states where it will not fail. For the goal action, we will have corresponding symbolic goal action whose precondition corresponds to potential states in $S_G$ and the effect is a goal predicate. The initial state consists of only the predicate corresponding to the state $I$ and the goal corresponds to the goal predicate. Now we can convert the actions with conditional effects to ones with no conditional effect (cf. [30]). Now the action mapping $\mathcal{C}^{-1}$, will map each of these new actions to the original MDP action from which it was defined. Now a plan is only valid in this model, if there exists a sequence of transitions from initial state to goal with non-zero probability. Similarly, for every valid trace there must exist a valid plan where each MDP action could be replaced by one of the potential symbolic actions that maps to it.

Next we will talk about the optimism of the initial model estimate

**Proposition 3** $\mathcal{M}_0^{\mathcal{C}} = \langle F_\downarrow^{\mathcal{C}}, A_\downarrow^{\mathcal{C}}, I^{\mathcal{C}}, G^{\mathcal{C}} \rangle$. *More formally, every action* $a \in A_\downarrow^{\mathcal{C}}$ *will be defined as follows:* $a = \langle pre_0^a, add_0^a, del_0^a \rangle$, *where* $pre_0^a = del_0^a = \emptyset$ *and* $add_0^a = F_\downarrow^{\mathcal{C}}$ *is optimistic for any MDP model such that there exists a mapping* $\mathcal{C}$ *from MDP state to symbolic states and a function* $\mathcal{C}^{-1}$ *mapping symbolic actions to MDP actions.*

This can be easily shown by the fact that every possible action sequence is a possible plan here.

Moving onto the update rule.

**Proposition 4** *Update rule as presented in Section 4.1, will only result in an optimistic representation.*

**Proof Sketch:**  The important point to note is that at any point, the update rule is only applied to an optimistic representation. So, in order for it to result in a non-optimistic model, it must have removed a plan corresponding to a valid trace. Given our initial construction of $\mathcal{M}_0^{\mathcal{C}}$, we always ensure that in $\mathcal{M}_0^{\mathcal{C}}$ the execution of an action $a$ at a state $\mathcal{C}(s)$ will result in a symbolic state that is a superset of $\mathcal{C}(s')$, where $T(s, a, s') = 1$. Note that an application of an update rule will only extend the precondition if the corresponding MDP action fails and the preconditions are extended to exclude only the current state (though the list of excluded state, action pairs grows as the number of failed samples grows). Additionally, the effect is changed only to disallow impossible transitions. Since the transitions are deterministic, only one sample is needed to determine that no other transitions are possible from that state and action. This means that the above property (the fact that the resultant

symbolic state will be superset) will be preserved through updates. Which in turn means that any plan that previously corresponded to a valid trace can become invalid.

Now coming to the theorem

**Theorem 1** *Algorithm 1 (as referenced in the main paper) will (a) terminate in a finite number of steps and (b) identify a path to a goal (provided one exists); as long as the diverse planner used is complete (i.e., it will return a non-empty plan set as long as there exists a valid plan).*

**Proof Sketch** The validity of this theorem follows from the fact that the update rule will remove any plan that doesn't correspond to a valid trace from consideration again. If the planner is complete then it will effectively iterate over all possible plans. Eventually finding one that corresponds to a path that goes to the goal. This is guaranteed to exit in finite steps, as the set of non-redundant plans is guaranteed to be finite when the state space is finite.

Now revisiting Proposition 1

**Proposition 1** *Let $\bar{A} = \langle a_1, ..., a_m \rangle$ be a set of actions marked as being instances of a single lifted action $a^\uparrow$. Then* **min_add** *must be a superset of add effects of $a^\uparrow$ and* **max_del** *a superset of deletes of $a^\uparrow$, where* **min_add** *and* **max_del** *are calculated for $\bar{A}$*

**Proof Sketch** The validity is trivial. The update rule makes sure that every effect estimate will be an optimistic estimate of the true ground action effects. In the case of add effects this estimate will be a superset and for delete effects it will be a subset. Thus, the lifted representation of each set must correspond to optimistic estimates of the true lifted representation of the effects.

## C   Implementation Details

All experiments were on a laptop running Mac OS v 11.06, with 2 GHz Quad-Core Intel Core i5 and 16 GB 3733 MHz LPDDR4X. We did not use CUDA in any of the experiments. For the planner, we used the FI-diverse-agl planner provided as part of the forbid iterative planner. As discussed we generated 10 plans in every planing query. The search was given a maximum threshold of 1000 iterations, but we never reached that limit given our time limit. We stop an action from being considered if it fails 10 times in a row. We will update this upperbound on number of failures if the planner returns empty plan at any point. Since we found out that the planner was slowed down by the introduction of disjunctive preconditions, we replaced the disjunctions with a set of actions (this is an equivalent compilation popular within planning). To control the growth of the precondition, we introduce an upper bound on its size, set to 10 in our experiment. Note that the true size of the preconditions in all instances we consider here is significantly smaller than our bound. We could make the bound adaptive to a domain, but we do not expect it to make any significant difference. For all the RL baselines we used a discount factor of $\gamma$. For Q learning and R max, we used a maximum of 1000000 episodes with 200 steps per episode. For exploration, the $\epsilon$ and decay rates were set as the same as the one used by SimpleRL experiment scripts. For PPO, we used the same default values used by [26]. The environment names for the two problems we tested in minigrid where where, MazeRooms-8by8-DoorKey-v0 and MazeRooms-2by2-TwoKeys-v0. While creating the PDDL model for minigrid we combined the turn actions with the other actions (move, pickup, drop, etc.), to avoid potential conditional effects.

## D   Additional Experiments

### D.1   Comparison with a Symbolic Baseline

To see how our method compares against other methods for symbolic model acquisition, we look at how many samples are required by a popular model learning method (cf. [19]) to generate a model that can produce a goal reaching plan. Since we don't have access to a plan library, we will generate one through random walks on our simulator. We focused on the Blocksworld domain, and for each of the five problems, we look at the number of samples required to generate a model that allows a potential plan to the goal. Since the method generates a pessimistic approximation of the model, any plan generated by the model is guaranteed to be valid and thus the method no longer requires the use of diverse planners to generate potential plans. We placed an upper bound of 600000 on the number of samples originally collected from the simulator (this is nearly six times larger than the number

of samples required by our method). Out of the five problems, we found that only the method was only able to learn a model capable of generating a valid plan in the case of the first problem instance. Even for that problem, we found that the method took on average 548287.8 samples.

## D.2   Kitchen Domain

As an additional experiment, we tested our method on the symbolic part of the Kitchen Domain [36]. We tested our method on the domain by creating a symbolic simulator that uses the descriptions provided in the appendix of the paper (specifically the inter-dependencies listed in Figure 5). The purely symbolic domain consisted of one action for each high-level goal possible and the preconditions were built based on the relationships described in the paper. The exact domain consisted of 15 predictions and 13 actions. The goal was the same as the one described in the paper (i.e., both banana and cabbage is cooked, they are placed on plates and the plates are served). The lifted version of our method was able to identify a valid plan in 61.46 sec using 24727.2 time steps (averaged across five runs) and the non lifted version took 393.57 secs and used 140681 samples (again averaged across five runs). Now executing the plans in the true simulator would require an additional component an additional step to drive the simulated robot to achieve each of these subgoal. However, as discussed in the paper, we can do that by using a motion level planner (like an RRT based planner).

## E   Assumptions Made

Here we explicitly mention all the theoretical assumptions we have made in the paper and how to relax them

**Model dynamics:**

Deterministic model – our primary formulation and evaluation focus on deterministic domains. However, as discussed in the future works section, we can directly apply our method to stochastic environments by creating symbolic models that correspond to so-called *all outcome determinizations* of the model [39].

**Observability:**

We assume that the environment is fully observable. However, previous work have looked at how partial observability can be compiled into classical planning model. For optimistic estimates, we can make further simplification to assume that all unobservable facts are true, thus allowing us to directly apply our methods in such settings.

**Finite state and action space:**

We assume that the underlying state and action space is finite and thus can be represented exactly using a finite symbolic models. For cases where this is not true, we can still employ our symbolic model to capture an abstraction of the true state and action space.

**Symbolic observations:**

We assume that noise-free classifiers for each symbolic fluent are given. Previous work [32] have looked at the problem of learning such classifiers, which we can directly use in our scenario. If the classifiers are noisy, this corresponds to a special case of partial observability. As discussed above, we therefore can extend our model to handle noisy classifiers.

## F   Overall Learning System

Here we provide an overview of the overall learning process. Figure 2 presents a diagrammatic representation of the learning process. As discussed before, we start with a trivially optimistic representation of the model. We then use a diverse planner to potentially generate possible plans from that domain, which are then tested on the simulators. The experiences generated from the simulator are then used to update and refine our optimistic representation. Once a successful plan is identified, this information is used to initialize the RL algorithm. For our experiments, we focused on Q learning; as such, the plans were used to initialize the Q values with more informative estimates. It is worth noticing that the problem of learning a useful refinement of an optimistic model involves solving an additional exploration-exploitation problem. Specifically, this involves identifying what actions

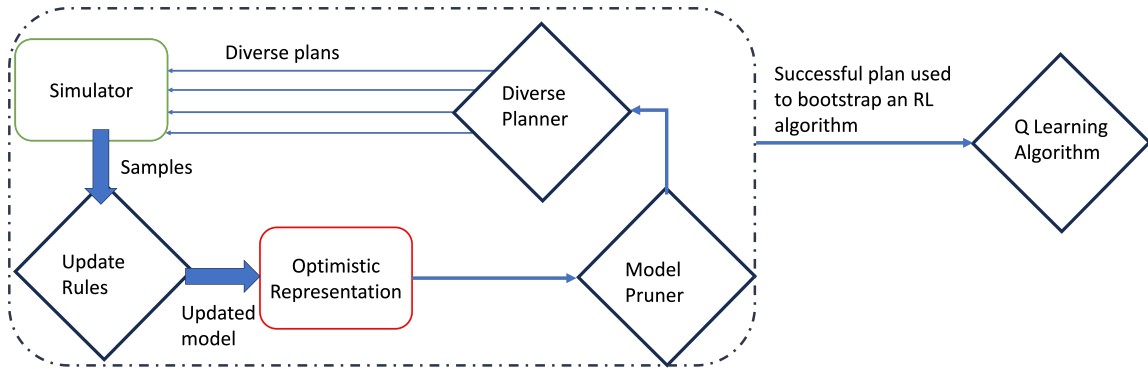

Figure 2: A visualization of the overall learning process

the planner should try to include in the plans. While the use of diverse planners already provides us with mechanisms to promote exploration, given the optimistic nature of the model estimates, there is always a possibility that the planner will try to use actions that have not yet been successfully executed (thereby updating its effect). In our current implementation, we employ a queuing strategy to prevent the planner from retrying the same actions too many times.

