# OpenReview forum: "Optimistic Exploration in Reinforcement Learning Using Symbolic Model Estimates"
_NeurIPS.cc/2023/Conference — NeurIPS 2023 poster_

### Official Review · Reviewer_MX4H · 2023-06-13

**Soundness:** 2 fair
**Presentation:** 4 excellent
**Contribution:** 3 good
**Rating:** 6
**Confidence:** 4

**Summary:**

This paper proposes a new method for optimistic exploration in sparse reward settings for reinforcement learning problems.  The core of this method is learning optimistic symbolic approximations of the underlying world model. Such optimistic symbolic models can then be used with a diverse planner to generate plans for explorations, which could ultimately facilitates learning. The proposed method is tested in four benchmark domains with grid world type environments. The proposed method is compared with three baseline methods, including two exploration methods and one hierarchical RL method from prior work. With the same computational time limit, the proposed method is shown to outperform the baselines in number of problems solved (goal reached). The proposed method is also shown to be able to effectively leverage human input through lifted representations in the symbolic model.


**Strengths:**

- The proposed method of learning optimistic symbolic models and utilizing them for exploration is technically interesting and novel.
- The paper is well-written and easy to follow.
- Exploration in RL is a long-standing and important problem. Leveraging symbolic methods in RL is a promising direction. This paper makes an important step in this direction.
- The experimental setting is sound: the proposed method is evaluated on four traditional benchmark domains and compared against three existing methods from prior work.

**Weaknesses:**

For experimental results, the author states that both R-max and SMDP timed out in all test instances under the time limit set in the paper, and therefore no result was reported in the paper on these two baselines. It would be helpful to report at least some results on these two baselines, otherwise it is not clear how the proposed method compare with these two baselines in different aspects. As an example, the author could set a very high time limit such that R-max and SMDP can finish running, then report the number of test cases solved as well as the average computational time used, and compare the proposed method with these two baselines in these statistics.


**Questions:**

Given a higher time limit, how does the proposed method compare with R-max and SMDP baselines?


**Limitations:**

I did not see a discussion on the limitations. It might be helpful to discuss what is assumed to be provided for the symbolic model for the proposed method to work, and what can we do if these elements are not provided.

---

> ### Author Rebuttal · Authors · 2023-08-09
>
> We thank the reviewers for their suggestions. Since most of the baseline methods considered, including Rmax and epsilon greedy visit all states in the limit, they will eventually find a goal state after a number of iterations. For planning tasks considered here, this number however can (and in many cases will) be prohibitively high. Having said that, we will make sure to update the paper with results for higher time limits.
>
> Limitations: We have included a discussion of the specific assumptions made in section 5 of the pdf included as part of the appendix.

---

### Official Review · Reviewer_6i1K · 2023-06-30

**Soundness:** 2 fair
**Presentation:** 2 fair
**Contribution:** 2 fair
**Rating:** 4
**Confidence:** 4

**Summary:**

This paper studies reinforcement learning in deterministic MDPs in which a symbolic representation of the state space is assumed to exist. The method proposed in this paper learns a symbolic model of the environment, and it uses it to speed up the reinforcement learning process. The algorithm starts with an "optimistic" model in which all actions lead to the desirable goal state. The role of the algorithm is to refine the actions to make them more realistic by discovering the preconditions and the effects of actions. A few mathematical assertions are provided in the paper, and the empirical results compare the new method with the Q-learning algorithm.

**Strengths:**

* The paper studies an important problem of enhancing reinforcement learning through symbolic models from symbolic PPDDL-like planning
* The contributions of the paper that are related  to symbolic planning are strong
* The fact that the paper uses lifted inference is a great plus
* The paper presents both empirical results and explains the method using theoretical arguments

**Weaknesses:**

* As said above, the contribution to the symbolic planning literature is clear and seems strong, but the link with reinforcement learning is unclear. For example, it is not clear to me how exactly the model being learned by Algorithm 1 interacts with Q-learning. A pseudo-code of a complete Q-learning algorithms with the new stuff would be very useful. Lines 285-294 are not sufficient to explain this integration.

* Since the MDP is deterministic, there is no need for a full RL policy since the solution can be a single trajectory. This means that Q-learning may not be required. Perhaps it would be more appropriate to integrate and compare this method with LRTDP, which would have convenient properties in deterministic MDPs.

* The authors were careful to acknowledge in the introduction that their models are optimistic with respect to the underlying transition function. But, the authors should also note that the word "optimistic" has a very special meaning in RL and in heuristics search or symbolic planning in general. Optimistic is often used as a synonym for admissible, and unfortunately the symbolic models studied in this paper are not optimistic nor admissible with respect to the expected rewards optimized by Q-learning. Note that one may have a very low-reward action that leads to the goal in 1 step, but such an action should never be chosen, when a longer trajectory can reach the goal at a much higher reward. The current approach will miss that. For that reason, I feel that the use of the word "optimistic" may confuse the readers in the future. Perhaps the authors could think about this. The authors should note that the R-max algorithm that the authors cited led to many other ways of using optimism in RL, e.g. https://icml.cc/Conferences/2010/papers/546.pdf or https://www.ifaamas.org/Proceedings/aamas2012/papers/1C_1.pdf or https://www.ifaamas.org/Proceedings/aamas2010/pdf/01%20Full%20Papers/06_06_FP_0313.pdf. The use of a diverse planner is good, but the point of the methods mentioned above is that they can do much better than a diferse planner which can go in a wrong direction and explore useless states.

* It seems to me that the authors could survey the literature on symbolic planning in deterministic domains when the PDDL operators are learned from data. I am sure that there must exist papers in which this problem was studied. I think that some literature review on this would be useful in this paper.

**Questions:**

* What is the difference between online and offline RL in lines 24-25?
* The sentence in line 81 criticizes [22], but the point made in that sentence is not clear to me.
* Def. 1 is confusing to me. If the action sequence has k actions in the full symbolic model, the optimistic sequence may have only one action which would lead to the goal straight away. I am not sure why both sequences have the same length in the definition. Also, the definition itself is cryptic to me, and only the subsequent paragraph managed to explain the idea of this particular optimism to me.

**Limitations:**

* The discount factor gamma should be part of the MDP definition. This is part of the environment, not the algorithm. This would also make the paper clearer. For a moment, I thought that discount actor is 1, but then given only the goal reward, the length of the trajectory would not matter. But, line 104 indicates that gamma is 1 indeed because "any" trajectory is seen as optimal. There are some conflicting statements in the paper.

* There are a few small typos and grammatical errors in various places. E.g. the exponent in line 163 or PDDL is written as pddl on p. 7.

* Line 420: how where the Q-values initialized?

* Appendix mentions Fig. 5, but the figure is missing.

---

> ### Author Rebuttal · Authors · 2023-08-09
>
> We thank the reviewer for the comments, and we will make sure to include a pseudo code for the updated Q-learning algorithm. We will make sure to include a more detailed discussion of the existing work in the space of learning symbolic models. We would also like to point out that the supplementary file includes a comparison to a popular symbolic model learning algorithm.
>
> Weaknesses:
>
> Contributions to RL - In so far as planning is the problem of using fully defined models to generate a course of action and RL learning the best course of action from experience, we are solving an RL problem. Additionally, we show theoretically that assuming a symbolic structure provides a number of advantages. We empirically show that our method outperforms a state-of-the-art hierarchical reinforcement learning approach on popular RL benchmarks.
>
> LRTDP -- The LRTDP algorithm is intended for non-deterministic domains. On deterministic domains, it will perform many unnecessary computations. We are not aware of any evidence of why this particular algorithm would work better than the popular Q-learning. The more efficient planning algorithms that do not require action models would be Greedy Best-First Search (GBFS) with weak evaluators that do not require the action model, like goal-count heuristic or Best-First Width Search. However, these algorithms can benefit greatly from the knowledge of action models, and that is what most symbolic planners are doing, including the diverse planners used here. It is a very well-known fact within the planning community that these search algorithms perform much better when equipped with evaluators obtained from action models and therefore, we did not feel the need to perform such an experiment.
>
> Optimism -- Please note that in this paper, we are trying to formalize and learn an optimistic representation of the underlying model. As discussed in Definition 1, we define this optimism in terms of the number of traces allowed (i.e., has non-zero probability in the model). In the paper, we were careful to frame all our claims of optimality in terms of learning an optimal representation and not necessarily in terms of optimal policy identified on top of the learned model. We will make sure to update the paper to clarify any such confusion.
>
> Further enhancements to the method: We really appreciate the links to the works that extend Rmax. Given the fact that the nature of the optimistic estimation process is drastically different from Rmax, it is unclear how we could map the intuitions from these works to our current approach. However, we agree this would make for an interesting next step for our work.
>
>
> Questions:
>
> Offline vs. online RL in the context of symbolic model learning: The methods we referred to as belonging to offline RL assume access to a set of traces. On the other hand, the method we employ directly interacts with the environment (or a simulator).
>
> Definition 1 - Please note that definition one doesn’t limit itself to just the optimal sequence possible under the model but rather considers all sequences with a non-zero probability. As such, the set of all traces (i.e., state action sequences with non-zero probability) under the learned approximation is guaranteed to be a superset of traces possible under the original model. We will work on updating the definition to make it easier to understand.
>
> Automatic synthesis of symbols: Note that we are building models on top of symbols (predicates, objects, actions, etc.) that are provided by the user. As such, one would expect the user to be able to make direct sense of the learned model descriptions, or at the very least, we could expect these models to be used as inputs to existing explanation generation methods for symbolic models. However, with methods that directly synthesize the symbols as well, there is no guarantee that people can make sense of the learned model descriptions.
>
> Discount Factor - Thank you for pointing out this typo; we will make sure to include the discount factor in the model definition. We would like to note that our method can support a discount factor of `1’. Given our focus on deterministic models, any policy that only generates finite paths to the goal will be considered optimal. Note that any policy that doesn’t end support a path to a goal state or includes a loop will not be optimal (as the loop will never terminate). We will update the discussion around line 104 to clarify that this is only true when the discount factor is 1.
>
> Q Value Initialization: They were initialized to zero.

---

> > ### Comment · Reviewer_6i1K · 2023-08-20
> > **Your answers read**
> >
> > Your answers clarified a few things to me, and the other reviews are helpful to see your work in a better light. As a result, I will increase my score sightly.

---

> > > ### Author Response · Authors · 2023-08-21
> > > **Response to Reviewer**
> > >
> > > We thank the reviewer for taking the time to go over our response and update the score. If you have any other questions about our method, evaluation, or the significance of the contributions, we would be more than happy to provide additional details.

---

### Official Review · Reviewer_S1WD · 2023-07-05

**Soundness:** 4 excellent
**Presentation:** 3 good
**Contribution:** 3 good
**Rating:** 7
**Confidence:** 3

**Summary:**

This paper follows a long line of recent work on trying to combine RL with symbolic reasoning. The main motivation for doing so in this paper is to improve exploration on sparse reward, goal-oriented tasks, similar to the "taskable RL" setting of Illanes et al. A major limitation of existing work is the need for a human expert to provide a symbolic model upfront. This paper aims to relax that assumption by removing the need to specify the transition logic (in RL parlance). The human still needs to specify how the states are represented symbolically, but an algorithm takes care of learning the model. In a nutshell, the way the algorithm works is by starting with the most permissive model possible, then iteratively asking the planner for a solution, attempting to execute it, and updating the model based on the sampled experience. The preconditions, additions and deletions are updated according to the least restrictive explanation of what just occurred, ensuring that the model remains "optimistic", i.e., any possible trace in the MDP remains possible under the planning model. This guarantees that the algorithm will eventually find a solution, provided one exists. The evaluation shows that approach learns quickly relative to some standard RL exploration strategies, and also shows that the approach can effectively bootstrap lifted representations learned from previous task instances when tackling new tasks.

**Strengths:**

Overall, I thought this was a very interesting paper, based on an intuitive idea. The approach seems more similar to how humans tackle long-term, sparse reward tasks compared to the standard RL approach of intrinsic rewards. Generally, we humans don't possess a complete model of the environment, so we come up with a *plausible* plan, try it out, then update our models based on what happened. It always amazes me how complicated ideas like this sound when expressed in planning logic (I'm more of an RL person), but surprisingly I was able to follow most of the core technical details. Largely this is because the paper is very well-written -- unlike a lot of papers I've reviewed recently, this one is actually self-contained, with the authors going to great care to explain the preliminaries and problem setting properly. This was much appreciated as someone who needs to be reminded of all the planning lingo. The coverage of related work is good (the authors come across as being very knowledgeable across many subfields) as a far as I know the approach is original. The experiments are mostly compelling (with some caveats noted below), and I thought the curriculum learning experiment (Section 5.3) was particularly cool!

**Weaknesses:**

The main weakness of the paper is one that's already acknowledged by the authors; namely, the assumption that the environment is deterministic. This limits the applicability of the method quite a lot, and I'm not sold from the brief explanation in Section 6 that the method would work well if directly applied to stochastic environments. Moreover, assuming determinism greatly simplifies some aspects of the decision-making problem. This is partly discussed on page 3, but there are a lot of things that one can do to speed up learning that aren't mentioned. For example, rather than applying Q-learning with 1-step updates, you can calculate all possible n-step return estimates and backup from the largest one. Also, until a reward is found, all action-values will be tied at 0, so an e-greedy policy is actually just a uniform random policy until that point (assuming random tiebreaking), which seems woefully inefficient if you know that the environment is deterministic. It's trivial to learn the transition function in the deterministic setting, so you could adopt a simple strategy like maintaining a list of visited states with untried actions, and actively work towards ticking off that list. I'm not super familiar with R-MAX, but I understand that it's designed for stochastic environments, which also puts it at a disadvantage assumption-wise. I'm not suggesting that the authors derive a brand-new RL algorithm that perfectly exploits determinism just to baseline against (although I'd be very surprised if no-one has tackled this problem before), but I'd be more convinced by the experiments if the baselines were at least slightly optimised for this setting.

My only other (minor) criticism of the paper is that it clearly ran into issues with the NeurIPS page limit. Much of the writing has been packed into walls of text to address this, and Figure 1(b) has been compressed to the point where it's very hard to read. While there's little you can do about this (I can't think of anything major that's cuttable), I have to admit I groaned when I opened the paper and saw the formatting. I'd definitely recommend releasing a nicer, less compressed version of the paper as a supplemental sometime in the future, since I think the current format will put off some readers.

**Questions:**

Could you please go into a bit more detail about what the main challenges would be in extending your approach to stochastic environments? For example, do you envisage learning transition probabilities and using a probabilistic planner, or would you consider something like FOND planning?

**Limitations:**

As mentioned in my comments above, I think the paper could do a better job of acknowledging the drawbacks of assuming a deterministic environment. This is very downplayed at the moment. It's promised early on that "In Section 6, we will see how we can also apply our methods directly in settings with stochastic dynamics", but then all that's actually given is a one-liner.

I don't see any potential ethical concerns with this work.

---

> ### Author Rebuttal · Authors · 2023-08-09
>
> We thank the reviewer for their comments; we will make sure to expand our discussion on how the method could be applied in stochastic settings.
>
> Weakness/Questions:
>
> Ease of learning models in deterministic settings: We would like to point out that a brute force-based method will not scale well in the domains we consider, given the sheer state space sizes involved. A good demonstration of this fact is the symbolic model learning baseline provided in the appendix main pdf (Section 3), which relies on random walks to generate different traces.
>
> Extending the method to stochastic settings: Please note that central notions of the optimism considered in the paper carry over to stochastic settings. We can still start with a model with empty preconditions and add effects that contain all add effects. Now the challenge here is, of course, how does one refine the effects of this model when there are multiple effects possible. As alluded to in the paper, the conceptually most straightforward approach would be to treat each qualitatively different outcome one comes across as a new action and introduce it into the model. One can’t remove the old action copies directly as one is unsure if there might still be a yet undiscovered outcome of that action, which may produce that effect. However, our internal action prioritization system can be updated to ensure that the planner will not try to rely on those outcomes after a certain number of trials have passed. This would, in theory, correspond to learning effectively a determinization of the original model. Of course, one could combine the outcomes together to form a FOND-like representation of the model, and we could even associate probability estimates with each observed outcome.

---

> > ### Comment · Reviewer_S1WD · 2023-08-19
> >
> > Thanks for your response. I'm just confirming that I've read the other reviews and rebuttals, and I still think the paper should be accepted. While the paper is a little bit dense, the subject matter is very technical and makes this somewhat unavoidable. Reviewer 6i1K also has some concerns about the deterministic MDP assumption, but for me these aren't enough to sink the paper. It's very interesting work in a novel direction, and I think it's fair enough to leave this weakness for future work, since there's already a lot of contribution here. I'm not concerned by reviewer nfft's review, because it seems like they just didn't "get" the work, whereas I think many readers will. (Even the other negative reviewer, 6i1K, seemed to understand the core idea pretty well and could see some clear strengths.)

---

> > > ### Author Response · Authors · 2023-08-20
> > > **Reply to official comment**
> > >
> > > Thank you for the vote of confidence. We appreciate the time invested in reading our response and other reviews, and we especially appreciate your support. We believe that an opportunity to present our paper at a general machine learning conference of such a caliber would allow us to highlight the advantages of using the tools and ideas developed originally in the context of symbolic planning. If you believe that the paper deserves a higher score, we would be grateful if you upgraded your evaluation.

---

> > > > ### Comment · Reviewer_S1WD · 2023-08-20
> > > >
> > > > Fair enough, I am clearly saying that the paper should be accepted, so I have upgraded my score to a 7.

---

### Official Review · Reviewer_nfft · 2023-07-07

**Soundness:** 2 fair
**Presentation:** 1 poor
**Contribution:** 2 fair
**Rating:** 3
**Confidence:** 3

**Summary:**

The authors present a framework for learning symbolic representation of an underlying MDP that leverages key concepts of the planning community.
Specifically, they use the PDDL formalism to represent a planning problem that's acquired through agent interactions with an MDP. The authors show that this approach can lead to higher success rate in underlying planning problems that simply using standard Q-learning.

**Strengths:**


- The results show that the proposed method is able to learn a model that can be leverage by the planning agent. The authors show that they are able to learn a symbolic representation for the underlying MDP from the agent experience.

**Weaknesses:**


- The main weakness of this paper is its presentation and writing style. I found the text really confusing as it seems like the text is not self-contained. For example, line 155 "denote grounded instance by replacing the parameter with an object list", what objects? where is the list coming from?
in line 178, "... learn a  binary classifier that tests whether a ground truth predicate may be true in a given MDP state..." What do you mean it is learned by collecting positive and negative examples for each ground concept?
Algorithm 1, what are the DiversePlanner, UpdateModel, and PuneModel functions?

- It is not clear where the exploration is taking place. The title would lead me to think that it would be a central part of the paper, but in the main text, it appears to be just a comment on the use of the planners.

**Questions:**

- As a suggestion, it would have been extremely helpful to have a working example throughout the text or some visualization to explain concretely what the method is doing. What exactly is being collected to learn the binary classifier? What are the objects replacing the parameters?


- What was the actual planer used in this evaluation? The text mentions several times "diverse planners", but what was actually used?

**Limitations:**

No clear mention of limitations, no concerns on societal issues.

---

> ### Author Rebuttal · Authors · 2023-08-09
>
> We thank the reviewer for their comments; we will work on having a running example that better illustrates the various points made in the paper.
>
> Questions:
> Binary classifier: We are using handcrafted classifiers in the experiments. As such we didn’t need to collect positive or negative examples. Our methods are completely agnostic of how the predicates are obtained. The specific passage the reviewer mentioned points to the fact that earlier works have shown that such predicates can automatically be learned from examples. To clarify, in this work, we assume the predicates to be provided (either hand-crafted or learned separately). Our methods are completely agnostic of how the predicates are obtained. The specific passage the reviewer mentioned points to the fact that earlier works have shown that such predicates can automatically be learned from examples.
>
> Diverse Planner: We used FI-diverse-agl planner provided as part of the ForbidIterative planner. The details are provided (along with other implementation details) in Section 2 of the main file in the supplementary folder.
>
>
> Weaknesses:
>
> Object: The notion of objects, as used in this paper, corresponds to a concept used within relational representations. This representation scheme makes the ontological commitment that the world can be represented as a set of objects and relationships between the objects. In theory, an object could be anything. However, in practice, they usually correspond to what people would consider to be objects. For example, in the case of the blocksworld domain described in the paper, the objects would be the various blocks that will be stacked on top of one another. The notation for objects is presented in lines 127-128.
>
> Predicates: Please see the answer to question 1.
>
> UpdateModel, and PruneModel - These processes are described in sections 4.1 (lines 242 - 255) and 4.2 of the paper (267 - 280), respectively.
>
> Exploration: As with the case of Rmax, exploration here is automatically performed as part of the planning process. In our case, as the model estimate is refined through interaction, the planner automatically comes up with different paths it thinks will reach the goal. These plans are then tested on the simulator and then used to refine the model. The refined model then provides new plans. We provide a high-level discussion of the overall approach in lines 192 to 207.

---

### Decision · Program_Chairs · 2023-09-21

**Decision:**

Accept (poster)

**Comment:**

The reviewers are split on the decision here. The two main criticisms put forward center around the presentation/clarity of the work, and the assumption that the underlying MDP is deterministic. In contrast, the reviewers in favor of the work praise the paper's novelty, intuitive appeal, logical flow, and compelling experiments (with an emphasis on the novelty of the work).

Following some discussion, one of the reviewers has made a compelling case for the paper to be accepted in light of the paper's strengths, while agreeing that the primary limitation of the work is the assumption that the MDP is deterministic. The reviewer noted that this limitation should not prevent the paper from publication as the resulting problem is still both difficult and interesting. Instead, it is reasonable to treat the relaxation of this assumption as an opportunity for future work, which I tend to agree with.

Overall, in light of the above discussion (and absent any major critiques of the work that should block publication), I recommend acceptance.